# Functions of Plant Phytochrome Signaling Pathways in Adaptation to Diverse Stresses

**DOI:** 10.3390/ijms241713201

**Published:** 2023-08-25

**Authors:** Xue Qiu, Guanghua Sun, Fen Liu, Weiming Hu

**Affiliations:** 1Lushan Botanical Garden, Jiangxi Province and Chinese Academy of Sciences, Jiujiang 332000, China; nc405600210004@outlook.com (X.Q.); sungh@lsbg.cn (G.S.); 2School of Life Sciences, Nanchang University, Nanchang 330031, China

**Keywords:** phytochrome, light signaling pathways, biotic stress, abiotic stress

## Abstract

Phytochromes are receptors for red light (R)/far-red light (FR), which are not only involved in regulating the growth and development of plants but also in mediated resistance to various stresses. Studies have revealed that phytochrome signaling pathways play a crucial role in enabling plants to cope with abiotic stresses such as high/low temperatures, drought, high-intensity light, and salinity. Phytochromes and their components in light signaling pathways can also respond to biotic stresses caused by insect pests and microbial pathogens, thereby inducing plant resistance against them. Given that, this paper reviews recent advances in understanding the mechanisms of action of phytochromes in plant resistance to adversity and discusses the importance of modulating the genes involved in phytochrome signaling pathways to coordinate plant growth, development, and stress responses.

## 1. Introduction

Plants perceive light signals through photoreceptors, among which phytochromes are primarily responsible for absorbing red (R) and far-red (FR) light [1,2]. Phytochromes regulate plant growth and development by interacting with intermediate factors such as PIFs (Phytochrome-interacting Factors), HFR1 (Long Hypocotyl in Far Red 1), COP1 (Constitutive Photomorphogenic 1), SPA1 (Suppressor of Photochrome A-105), and HY5 (Elongated Hypocotyl 5) [1,3,4,5,6]. Through reversible photoisomerization of phytochromobilin, phytochromes are divided to two forms: Pr (able to absorb red light) and Pfr (able to absorb far-red light) [3,7]. It is believed that Pfr is the active form, that interacts with other proteins both in the cytosol and inside the nucleus [7]. Phytochromes and their associated signaling pathways play crucial roles in various physiological processes of plants, including germination, de-etiolation, shade-avoidance syndrome (SAS), flowering (heading), as well as biotic/abiotic stress responses [6,8,9,10,11,12,13,14]. The growth and development of plants are subject to various abiotic stresses, including high temperature, drought, waterlogging/submergence, salinity, and low temperature, as well as biotic stress such as pathogens and pests (Figure 1) [7,15,16,17,18,19]. To cope with these challenges, plants regulate their physiological processes through hormone pathways involving abscisic acid (ABA), ethylene (ET), salicylic acid (SA) and jasmonic acid (JA) [20,21,22,23,24]. Light signaling pathways are intricately intertwined with plant hormone signaling pathways, and phytochromes enable plants to evade or endure stress hazards by participating directly or indirectly in hormone signaling pathways [20,23,25,26,27,28,29,30,31]. Janda et al. also mentioned that phyB and PIF4 play important roles in resistance to high and low temperatures, but PIF4 is more stable at high temperatures and degrades at low temperatures [32]. Moreover, phyB also plays an important role in regulating reactive oxygen species (ROS) production in response to heat, cold, high-intensity light, and bacterial infection, and ROS may be another key node in the interaction between light and temperature signaling pathways [32,33]. This paper primarily summarizes advances in phytochromes and their associated signaling pathways during plant response to biotic/abiotic stress, while also elucidating the molecular mechanisms of light-mediated stress responses in plants. It aims to optimize plant resistance through phytochrome-mediated pathways, thereby maintaining the desired physiological state and achieving better economic output.

## 2. Phytochrome Signaling Pathways

In the course of long-term evolution and selection, plants have preserved several crucial photoreceptors: phytochrome (phy), which primarily absorbs far-red light (FR) at 700–750 nm and red light (R) at 600–700 nm to mediate far-red and red light signals; cryptochrome (CRY), which mainly absorbs UV-A in the range of 320–400 nm and blue light (B) between 400 and 500 nm to mediate blue light and ultraviolet-A (UV-A) induced plant responses; phototropin (PHOTO), which absorbs both blue and ultraviolet spectra; the ZTL family that responds to blue light, as well as UVR8 that is sensitive to light around 280–315 nm [34,35,36,37,38,39,40]. Phytochromes are essential photoreceptors in the plant’s photoreceptor system, primarily responsible for detecting and responding to far-red and red light stimuli, and they play a critical role in regulating seed germination, SAS, photomorphogenesis, flowering, vernalization, etc. [7,10,41,42,43,44,45]. Phytochromes of *Arabidopsis thaliana* consist of five members (*PHYA*–*PHYE*), which can be classified into three subgroups based on phylogenetic analysis: *PHYA*, *PHYB/PHYE* and *PHYC* [41,46,47,48]. Phytochromes of gramineae, such as rice, corn, and wheat, solely comprise three subfamilies: *PHYA*, *PHYB* and *PHYC* [49,50,51,52,53].

Phytochromes are dimeric proteins consisting of two identical apoproteins covalently linked with phytochromobilin, which confers upon them the capacity to absorb far-red or red light [7,54,55]. Phytochromes undergo reversible photoconversion between activated state (Pfr) and inactive state (Pr) (Figure 1), which is mediated by reversible photoisomerization [2,7,56]. It is generally believed that phytochromes in the Pfr state have biological activity and can interact with other proteins in the nucleus or cytoplasm, participating in the regulation of light signal transduction pathways [2,3]. However, recently, some scholars have proposed that the Pr form of phytochromes in the nucleus may also have biological activity [57]. Phytochromes include the N-terminal photosensory domain and C-terminal dimerization domain. The PAS and GAF subdomains in the N-terminal form a core photosensory center, which contain bilin lyase activity and ligating chromophore to PAS domain (bacterial phytochrome) or GAF domain (plant phytochrome) [58,59,60]. The main functions of the C-terminal are for dimerization and nuclear localization [54,61]. It should be noted that the C-terminal of Arabidopsis phytochrome A (AtphyA) can provide dimerization ability, but cannot provide nuclear localization signals, thus its nuclear localization relies on the help of auxiliary factors such as FHY1 (Far-red Elongated Hypocotyl 1) and FHL (FHY1-like) [62,63].

Phytochromes are thought to regulate follow-up processes mainly by interacting with other proteins, which are dependent on intermediate factors in the phytochrome signaling pathway, and COP1, SPA1, PIFs, HFR1, HY5, etc., are important intermediates in the phytochrome signaling pathway, of which PIFs play an important role in the regulation of plant response to biotic/abiotic stress [3,4,5,7]. PIFs are negative regulators of photomorphogenesis, interacting through the APA (active phytochrome A-binding) or the APB (active Phytochrom B-binding) with phyA or phyB in the Pfr state, and achieving the ubiquitination and degradation of PIFs by 26S proteasomes [64,65,66,67]. PIF1-PIF8 all have an APB and can interact with phyB; however, the APA is only found in PIF1 and PIF3, meaning that only PIF1 and PIF3 can interact with phyA [4,68]. Plants’ absence PIFs (single mutant or multiple mutants) exhibit photomorphogenesis in dark conditions, while the quadruple mutants of PIFs, pif1 pif3 pif4 pif5 (pifq) exhibit photomorphogenesis with opening cotyledons and shortened hypocotyls in darkness [69,70]. However, not all PIFs exist as negative regulators, as PIF6 can promote photomorphogenesis in Arabidopsis under red light condition [71,72] and PIF8 could activate the expression of *BBX28* to control H_2_O_2_ levels and prolong petal senescence in roses [73].

Although PIFs were originally discovered in light signaling pathways and named as phytochrome-interacting factors, they can be involved in many signaling pathways and perform functions. PIF1, PIF3, PIF4, PIF5 and PIF7 can be involved in regulating biological clock-mediated plant growth [74,75,76,77,78,79]. PIF4, as a key regulator of thermomorphogenesis, promotes the elongation of hypocotyls through the adjustment of transcription and post-transcriptional stability in high temperature conditions; moreover, high temperatures can also promote the expression of *FT* (*Flowering Locus T*) through PIF4 to lead to earlier flowering [80,81,82,83]. In low-temperature signaling pathways, PIFs participate in low-temperature adaptation, growth and development through CBFs (C-repeat binding factors) [26,84,85]. In tomatoes, PIF3 can be involved in the biosynthesis of photo-dependent tocopherol [86]. PIF4 and PIF5 can also be involved in regulating leaf senescence and immune responses in plants [87,88,89]. PIF8 also involved in growth, regulation of ROS level, cold tolerance and powdery mildew resistance [73,90,91,92,93]. In addition, PIFs are also involved in the regulation of hormone signaling pathways such as GA, ABA, ET, BR, etc., by influencing the synthesis of hormones, regulating the expression of, or interacting with key factors of the hormone signaling pathways [77,90,94,95]. In summary, PIFs, a family of basic helix-loop-helix (bHLH) transcription factors, have many roles in photomorphogenesis, hormone signaling, and biotic and abiotic stress.

HY5 plays a role as a positivity regulator under various light conditions (far-red, red, blue, and UV), and its protein abundance also shows a positive correlation with the degree of photomorphogenesis [96,97,98,99]. HY5 is located downstream of the light signaling pathway and can bind to the promoter of light-regulated genes, regulating photomorphogenesis by upregulating or downregulating gene expression [100,101]. HY5 can also coordinate light, temperature, and hormone signaling pathways, balancing growth and development with low-temperature resistance [25]. The COP1-SPA1 E3 complex serves as the core negative regulatory factor in the light signaling pathway, targeting key light-signaling positive regulators for degradation [3,5]. In the nucleus, COP1 binds to HY5 through the WD40 domain, mediating the ubiquitination and degradation of HY5 [97,102]. However, HY5 achieves its stability and activation through the phosphorylation of its COP1-binding region [103]. The interaction and degradation process between COP1 and HY5 can serve as a “switch” in the development of Arabidopsis mediated by light, synergistically regulating light signal transduction [5,104]. Similarly, COP1 can also regulate plant abiotic stress tolerance through regulation of HY5 and other factors [105,106].

## 3. Phytochrome Signaling Pathways and Abiotic Stresses

### 3.1. Phytochrome Signaling in Adaptation to High Temperature

Temperature is an important environmental factor during plant growth, and higher temperatures reduce seed germination and increase the length of the petiole. The change in plant morphology caused by diverse temperatures is called thermomorphogenesis, which is an adaptive reaction of plants [107,108]. Severe high temperatures are often accompanied by extremely intense light exposure. So, what is the relationship between the adaptation of plants to high temperature and phytochrome signaling? Phytochrome A-E are well known as photoreceptors, but later studies found that increase in temperature can promote the transformation of phyB into Pr forms without relying on light, which directly verifies that phyB can exist not only as a photoreceptor but also as a thermosensor (factors of phytochrome signaling for regulating abiotic/biotic stress are listed in Table 1, which will not be repeated in the following) [109,110,111]. The absence of *PHYB* in Arabidopsis enhances thermal tolerance; the rate of leaf appearance accelerated by high temperatures is slowed down in the *phyB* mutant [112,113]. PhyB perceives high temperature and modulates the accumulation of chlorophyll and carotenoid in tomatoes [113]. PhyB also perceives shade signals, endowing Arabidopsis with heat resistance [114]. High temperature and phyB antagonistically coordinate seed germination, and the S-nitrosylation and degradation of HFR1 play an important role in high-temperature suppressing germination [115]. PIFs are central regulators of photomorphogenesis. In plants, the role of PIF4 is most pronounced when plants are exposed to high-temperature stress [64,81,116,117]. Under high-temperature stress, PIF4 is abundantly expressed in plants as a crossover protein between phytochrome signaling and temperature signaling to initiate a protective pathway in plants under high-temperature adversity and thus enhance heat tolerance [81,117,118]. In light conditions, phyB induces phosphorylation and degradation of BIN2-mediated PIF4, while high temperatures induce excessive phosphorylation of PIF4 which enhances the stability of PIF4 [119,120]. HEMERA is a key regulating factor in phyB-mediated photomorphogenesis, which interacts with PIF4, inducing thermomorphogenesis associated gene expression and PIF4 accumulation, and participating in the thermomorphogenesis of plants [121]. PIF4 can directly interact with the promoters of *NAC019* (*NAC Domain Containing Protein 019*) and *IAA29 (Indole-3-Acetic Acid Inducible 29)*; therefore, PIF4/5 can activate SAG113 (Senescence-associated Gene 29) and NAC019, while repressing *IAA29* and *CBF2* expression to complete the regulation of leaf senescence under high-temperature stress in Arabidopsis [122,123]. Under high temperatures (28 °C), TCP17 bind directly to the promoter of *PIF4* and promote its transcription [124]. In turn, PIF4 binds to the promoter of the heat shock factor *HsfA2* for higher expression to maintain higher expression of heat-stress-related genes and enhance plant heat tolerance [30,125]. In addition to PIF4 and PIF5, PIF7 could also respond to high-temperature stress. Under high temperatures, plants can rapidly accumulate PIF7, and then induce transcription of *YUC8*, *YUC9*, *IAA19*, *IAA29*, etc., to initiate thermomorphogenesis [118,126]. HY5, the bZIP transcription factor, inhibits the expression of *PIF4* and competes for target genes with PIF4, participating in thermomorphogenesis [127,128]. However, high temperatures cause COP1 to enter the nucleus and promote the degradation of HY5 through the 26S proteasomes [105]. Red and blue light can cause phyB to initiate HsfA1-mediated expression of *APX2* (*Ascorbate Peroxidase 2*) to accelerate the removal of ROS under high-temperature stress [129]. In conclusion, phyB and PIFs play important roles in coping with high-temperature stress; phyB is a temperature sensor, and PIFs are central factors for crosstalk in adaptation to high temperatures (Figure 2).

### 3.2. Phytochrome Signaling in Adaptation to Low Temperature

Low temperature is not conducive to plant growth and development, severely inhibits life activities, and even causes structural damage to cells and tissues. Chilling and freezing damage all belong to low-temperature stress. Phytochrome-sensitive mutant *hp1* and phytochrome-deficient mutant *aur* exhibited different physiological, biochemical and molecular responses under chilling, which means phytochromes play a role under low-temperature stress [130]. It has been demonstrated that phyB, PIF3, PIF4, and PIF7, etc., have important roles in adaptation to low-temperature stress in Arabidopsis [84,131]. PIF3, PIF4, and PIF7 can negatively regulate *CBF* expression by binding to the promoter to reduce plant freezing resistance [84,131,132,133]. In addition, CBFs interact with PIF3 to attenuate the mutually assured destruction of PIF3 and phyB, and the cold-stabilized phyB positively regulates freezing tolerance by regulating growth-related and stress-responsive genes [26]. Low temperature inhibits the protein degradation of PIF3 mediated by the F-box proteins EBF1 (EIN3-BINDING F-BOX 1) and EBF2, while AtPIF3 regulates the ability to tolerate low temperatures by directly inhibiting expression of *CBFs* [84]. In tomatoes, phyA and phyB antagonistically regulate cold tolerance via ABA-dependent JA signaling [134,135,136]. Similarly, PIF4 can also bind to the promoters of *CBF1* and *GAI4* (*Gibberellic Acid Insensitive 4*) to activate their expression and then enhance cold tolerance [28,137]. PIF8 increases the expression of the *SOD* (*Superoxide Dismutase*) gene and the activity of SOD to reduce the superoxide anion (O_2_^−^) level to enhance cold tolerance in citrus [92]. Research has shown that HY5 is also a positive regulator of the cold signaling pathway, which induces the expression of *CAB1* (*chlorophyll A/B binding protein 1*) by combining Z-box and other cis-acting elements of its promoter, mediating the process of cooling domestication and enhancing cold resistance in plants [138]. Under low temperatures, HY5 can also directly target the ACE components in the promoter of the genes *BBX7 (B-BOX DOMAIN PROTEIN 7)* and *BBX8*, thereby altering their gene expression to integrate light and cold signaling pathways [139]. HY5 is also essential for cold tolerance by binding to promoters of *NCED6* and *GA2ox4* to reduce GA/ABA ratio in tomatoes [25,140]. At low temperatures, low R:FR promotes FHY3 (Far-red Elongated Hypocotyl3) accumulation, while the FHY3 interacts with HY5, and then enhances the accumulations of HY5, to improve the resistance to low temperatures by regulating the synthesis of ABA, the accumulation of inositol, the photoprotection pathway, and the CBF-mediated cold-resistant pathway [25,138,141,142,143]. When dealing with low temperatures, the phytochrome signaling pathway is actively involved in regulating the related genes and achieves effective control of low-temperature stress through the crosstalk with hormone signal pathways (Figure 3).

### 3.3. Phytochrome Signaling in Adaptation to Drought Stress

Water shortages are a major problem facing modern agriculture, which severely restricts crop growth and yield. Drought is harmful to plants, which compels plant to close stomata and accumulate ROS [144,145,146]. Under drought, plants launch a series of pathways to protect themselves, and ABA plays an indispensable role in drought resistance [24,147,148]. Numerous studies have shown that the phyB as well as PIFs can influence the content of ABA to regulate drought tolerance in plants [11,29,31,149,150,151,152]. PIFs can improve scavenging ability of photosystem I (PSI) and photosystem II (PSII) to ROS under drought stresses and increase ABA content to initiate the expression of drought-related genes for greater drought tolerance [23,24,29,150,153,154]. PIFs can also promote stomatal closure to reduce transpiration rate and enhance drought tolerance [11,155]. Drought inhibits the expression of *OsPIL1* and *OsPIL13*, while overexpression of *OsPIL1* can enhance the resistance of rice to drought [156,157]. *PHYB*-deficient mutants of rice improve plant drought tolerance by reducing leaf area and stomatal density [152]. In Arabidopsis, phyB contributes to acclimation to drought stress by enhancing ABA sensitivity though altering expression of *ABCG22*, *PYL5*, *RAB18* and *RD29A* [31]. In tomato, *phyA* and *phyB* mutants exhibited drought tolerance, but the mutant of rice phytochrome B (*osphyB*) negatively regulates tolerance to water deficiencies by controlling stomatal density and total leaf area [158,159]. In addition, under drought conditions, tomato DELLA regulates changes of ABA receptors by inhibiting the biosynthesis of GA, thereby increasing the sensitivity of stomata guard cells to ABA, causing the stomata to shut down prematurely, and regulating tomato resistance to drought, but low R:FR promotes phyB inactivation and DELLA degradation, reversing the positive regulatory role of phyB and DELLAs in plant resistance to drought [160,161]. Under high R:FR conditions, phyB can accelerate the consumption of water by increasing the density and index of stomata to adapt the plants to the high light [162]. However, the increased consumption of water caused by high light also serves as a compulsory signal, leading to an increase in the content of ABA and causing the stomata to close rapidly [31]. ABI5 is a key transcription factor in the ABA signaling pathway. HY5 and ABI5 can activate the expression of *ABI5*; the regulator BBX21 can interact with HY5 and then inhibit the activation of *ABI5*, thus integrating the light signaling pathway and the ABA signaling pathway [163]. FHY3/FAR1 (Far-red impaired response1) can also directly combine with the promoter of *ABI5* and promote its transcription, thereby regulating seed germination and the response to drought stress [164]. In summary, the phytochrome signaling pathway mainly enables the regulation of plant resistance to drought through the ABA pathway and the expression of drought-related genes (Figure 4).

### 3.4. Phytochrome Signaling in Adaptation to Salt Stress

Unreasonable irrigation and excessive application of fertilizer are more likely to lead to soil salinization, thereby affecting crop growth, development, yield and quality. Salinity damage to plants is mainly reflected in oxidative stress, osmotic stress, and ion homeostasis [18]. Under salt stress, phytochrome-interacting factor-like 14 (OsPIL14) promotes mesocotyl and root growth by directly binding to promoters of cell elongation-related genes and regulating their expression [165]. The accumulation of PIF4 proteins can induce the expression of salt-related genes such as *SAG29* and *ORESARA1*, giving plants the resistance to salt [166]. Salt stress also increased the stability of DELLA protein SLENDER RICE1 (SLR1), and through OsPIL14-SLR1 transcriptional module to fine-tune seedling growth [165,167]. PIF1 regulates gene expression to increase ABA and proline content to enhance salt tolerance, and PIF8 enhances scavenging of ROS by increasing water uptake, retention capacity and osmoregulatory capacity to reduce salinity damage [150,168]. FLS2 (Flagellin-sensitive 2) and RBOHD (oxidase homolog) can regulate the expression of *PIF4*, thereby regulating salt tolerance in Arabidopsis. HY5 modulates salt stress response by orchestrating transcription of *HsfA2* in Arabidopsis, and the *HY2* also acts as a NaCl signaling positive regulator during seed germination [169,170]. Red light increases the expression of genes involved in proline biosynthesis and metabolism, such as *P5CS1* (Pyrroline-5-carboxylate synthetase 1) and *PDH1* (Proline dehydrogenase 1), which promote large amounts of proline and thus increase salt resistance, and the biological process requires the participation of HY5 [171,172,173]. Tomato *phyB1* mutants lost salt tolerance under low R:FR, which suggests an important role of phytochrome B in mediating salt tolerance in plants under different ratios of R:FR [174]. However, *phyA*, *phyB*, and *phyAB* double mutants of *Nicotiana tobacum* all showed better salt tolerance compared to the wild type, which means that the *PHYA* and *PHYB* genes of tobacco negatively regulate salt resistance [175]. In Arabidopsis, phyA/phyB enhance SOS2-mediated PIF1/PIF3 phosphorylation and degradation, which can promote salt tolerance [176]. Both light signals and salt signals can regulate the nuclear import of COP1, and as an important inhibitor of the light signaling pathways, COP1 can negatively regulate the protein levels of HY5 and ABI5, thus promoting seed germination under salt stress [106]. In addition, the increased level of HY5 protein in the photomorphogenesis-related mutant *det1* (*de-etiolated 1*) promotes the expression of *ABI5,* thereby making *det1* more sensitive to salt, which significantly reduces the seed germination rate during salt stress [177]. When expressed in wheat, *AtHFR1* can improve osmotic stress tolerances caused by NaCl and PEG during seed germination [178]. In general, under salt stress, phytochromes, PIFs, HY5, etc., through ABA, ROS, etc., regulate growth and development in plants (Figure 5).

### 3.5. Phytochrome Signaling in Adaptation to High-Intensity Light

High-intensity light (HIL) is also one of the abiotic stresses commonly encountered during plant growth and development. HIL can have a significant impact on the photosynthetic system of plants, and phytochromes, as photoreceptors play a positive role in adapting to HIL and regulating the plant photosynthetic system. Red and blue light stimulate the synthesis of chlorophyll and coordinate the positioning of leaves and chloroplasts to optimize the utilization of light [179]. Changes in light signals induce differential accumulation of phytochromes, while HIL hinders the synthesis and accumulation of chlorophyll and carotenoids, thereby regulating the photosynthetic system of tea plants under high light conditions [180]. In tomato, *PHYB1* and *PHYB2* antagonistically regulate various aspects of photosynthesis [181]. The *phyA* mutant of tomato showed reduced photosynthetic activity of the excised chloroplasts and decreased biomass in adult plants [179]. In the mutants of the photoreceptor, the effects of HIL on photosystem II (PSII) activity are different from those of WT. The expression of *CHS* and *APX* genes in phytochrome mutants decreases, key enzyme and antioxidant activities are lower, and pigment content is reduced, resulting in lower resistance of mutants to HIL [182]. HIL reduced the maximum quantum yield (Fv/Fm) of PSII, PSII performance index (PI_ABS_), and photosynthetic and respiratory rates. The phyB mutant exhibited decreases in the expression of genes *CHS*, *HY5*, *APX1*, and *GPX* and decreases in the content of carotenoids and pigments that absorb ultraviolet light, resulting in a decrease in the photosynthetic activity of the *phyB* mutant [183]. Deletion of *DET1* (De-etiolated 1) leads to pigment accumulation and increased expression of the *CHS* and *HY5* genes, resulting in a greater adaptation to HIL [184]. In *phyB* mutants, ROS cannot accumulate in cells in response to excess light stress [33]. The ROS wave triggered by the excess light stress, as well as the excess light-stress-triggered local and systemic stomatal aperture closure responses, all depend on the function of phyB [185]. Based on the above, the phytochrome signaling pathways regulate plant tolerance to HIL stress by altering the expressions of genes and activities of the photosynthetic apparatus.

**Table 1 ijms-24-13201-t001:** Regulation of abiotic/biotic stress by phytochrome signaling pathways.

Stress	Factor	Function
High temperature	phyB	phyB is a temperature sensor. Temperature increase promotes the transformation of phyB into Pr, without relying on light [109,111]. Modulates the accumulation of chlorophyll and carotenoid under high-temperature [113]. Perceives shade signals, endowing Arabidopsis with heat resistance [114]. High temperature and phyB antagonistically coordinate seed germination [115]. Initiate HsfA1 mediated expression of the *APX2* to accelerate the removal of ROS [129].
	HFR1	S-nitrosylation and degradation of HFR1 play a role in high-temperature suppressing germination [115].
	PIFs	PIF4 abundantly expressed to initiate a protective pathway in plants under high temperature [81,117,118]. PIF4 Regulate the expression of *NAC019, IAA29*, *SAG113*, NAC019, *CBF2*, TCP17 and *HsfA2* [30,81,117,118,123,124].
		Under high temperature, plants rapidly accumulate PIF7, inducing transcription of *YUC8/9*, *IAA19/29* to initiate the thermomorphogenesis [118,126].
	COP1, HY5	High temperatures cause COP1 to enter nucleus and degrade HY5 [105].
Low temperature	PIFs	Regulate *CBF* expression to reduce plant freezing resistance [84,131,132,133]. increases the activity of SOD to enhance cold tolerance in citrus [92]
	PHYs	phyBPositively regulate freezing tolerance by growth-related and stress-responsive genes [26]. PhyA and phyB antagonistically regulate cold tolerance [134,135,136].
	HY5	Regulate expression of NCED6 and GA2ox4 to reduce GA/ABA ratio [25,140]. induces the expression of *CAB1* to mediate the process of cooling domestication and enhance cold resistance in plants [138]. Alter the expression of *BBX7* and *BBX8* to integrate light and cold signaling pathways [139].
	FHY3	FHY3 interacts with HY5, and then enhances the accumulations of HY5 and the synthesis of ABA [25,138,141,142,143]
Drought stress	PIFs	Improve scavenging ability to ROS under drought stresses [24]. Increase ABA content and initiate the expression of drought-related genes [23,24,29,153]. Promote stomatal closure [11,155].
	PHYs	Rice *PHYB*-deficient mutants reduce leaf area and stomatal density [152]. phyB can enhance ABA sensitivity though altering expression of *ABCG22*, *PYL5*, *RAB18* and *RD29A* [31]. In tomato, *phyA* and *phyB* mutants exhibited drought tolerance [158].
	HY5	HY5 can activate the expression of *ABI5* [163].
	FHY3/FAR1	FHY3/FAR1 can promote the transcription of *ABI5*, thereby regulating the response to drought stress [164]
Salt stress	PIFs	Under salt stress, OsPIL14 promotes mesocotyl and root growth, by regulating cell elongation-related genes [165]. PIF1 regulates gene expression to increase ABA and proline content [168]. PIF4 proteins can induce the expression *SAG29* and *ORESARA1*, giving plant the resistance to salt [166] PIF8 enhances scavenging of ROS by increased water uptake, retention and osmoregulatory capacity to reduce salinity damage [150].
	HY5	Modulate salt stress response by orchestrating transcription of *HsfA2* [169]. HY5 promotes the expression of *ABI5* [177].
	HY2	Act as a positive regulator during seed germination [170].
	COP1	COP1 can negatively regulate the protein levels of HY5 and ABI5, thus promoting seed germination under salt stress [106].
	HFR1	*AtHFR1* can improve osmotic-stress tolerances in wheat [178].
	PHYs	PHYB mediate salt tolerance under different ratio of R:FR [174]. *PHYA* and *PHYB* negatively regulate salt resistance [175].
High intensity light	PHYB	In tomato, *PHYB1* and *PHYB2* antagonistically regulate various aspects of photosynthesis [181] The *phyB* mutant resulted in a decrease in photosynthetic activity [183] In *phyB* mutant, ROS cannot accumulate in cells in response to excess light stress [33] The ROS wave triggered by the excess light stress, as well as the excess light-stress-triggered local and systemic stomatal aperture closure responses, all depend on the function of phyB [185].
	PHYA	The *phyA* mutant of tomato showed reduced photosynthetic activity of the excised chloroplasts and decreased biomass in adult plants [179].
	DET1	Deletion of *DET1* (De-etiolated 1) results in a greater adaptation to HIL [184].
Biotic stresses	PIF8	PIF8 Inhibits the expression of *WRKY42* and *ICS*, negatively regulating SA biosynthesis and powdery mildew resistance [63]
	FHY3/FAR1	Enhance the resistance to plant disease by regulating the concentration of SA [64].
	PHYs	*PHYB* Weaken the promotion of JA signaling by MYC2, and reduce the sensitivity to biotic stress [20,186,187]. *phyB* not only enhanced the inhibition of the JA signaling pathway but also reduced the density of trichomes, ultimately weakening defense [188]. phyA/B interfere with SA- or JA-mediated plant defense systems [189,190,191]. Mutation of *PHYB* enhances the resistance to sheath blight by increasing the uptake of NH4^+^ [14]. *osphyB* has a better resistance to brown planthopper under dim light [13].

## 4. Phytochrome Signaling in Adaptation to Biotic Stresses

Unlike animals, plants cannot move to escape the harms of biotic stresses. In the face of biotic stresses, plants can rapidly increase the transcription of chitinase-related genes and use chitinase to resist biological harm, and specialized antimicrobial metabolites can also be produced for specific pathogenic microbial injuries [192,193]. In addition, plants adjust hormones such as ET, SA, and JA to initiate relevant defense mechanisms [21,194]. Notably, studies have shown that phytochromes play a role not only in the exposure of plants to abiotic stresses but also in biotic stresses [195,196]. Light, perceived by phyB and other photoreceptors, helps plants focus their defensive arsenals on photosynthetically valuable leaves coping with biotic stresses [197] (Figure 6).

JA plays an important role in plant response to biotic stresses [198], and can rapidly accumulate within 30 min after being biologically nibbled to initiate plant defense mechanisms [16]. Studies have found that inactivation of phyB in plants can lower the level of DELLA protein, reducing of the DELLA-JAZ complex and causing more binding of JAZ10 to MYC2, weakening the promotion of the JA signaling pathway by MYC2, and reducing the sensitivity of plants to biotic stress [20,186,187]. In tomato, mutations of *phyB* not only enhanced the inhibition of the JA signaling pathway by JAZ-blocking proteins, but also reduced the density of trichomes on plant leaves and stems, ultimately weakening physical defense [188]. *phyAphyB28*, the *Nicotiana tabacum* mutant of phytochrome A/B, exhibit lower resistance to *Chilli veinal mottle virus* (ChiVMV), which means phyA and phyB positively regulated defense responses to ChiVMV infection and the process relies on the JA and SA defense pathways [189] (Figure 6).

SA has a very important role in plant defense, which can act as a signal to initiate plant defense and as an important component of systemic acquired resistance (SAR) in plants [21,22]. The phytochrome-induced signaling pathway interacts with the SA-mediated signal transduction route to influence the SAR against pathogens by regulating pathogenesis-related proteins [199,200] (Figure 6). *PHYA* and *PHYB* were shown to regulate the accumulation of SA and interfere with SA-mediated plant defense systems on the defense of *Cucumber mottle virus* (CMV) in *N. tabacum* [190,191]. Similarly, in *N. tabacum* NC89, it was verified that deletion of *PHYA* and *PHYB* could affect the expression of SA signaling related genes *NPR1*, *PR1* and *PR2* to reduce the resistance to ChiVMV [189]. Not only that, *PHYA*, *PHYB*, and *PHYC* are also found to regulate the SA and JA signaling pathways in rice, which are required for developmentally controlled resistance to *Magnaporthe grisea* [201]. In addition, *phyB* mutation can enhance the resistance to sheath blight (ShB) by increasing the uptake of NH4+ [14]. Dim light reduces rice resistance to the brown planthopper (BPH) relying on elevating ET biosynthesis and signaling in a phyB-dependent manner, which means mutations of *osphyB* have better resistance to the BPH under dim light [13]. PIF8 inhibits the expression of *WRKY42* and *ICS* (*Isochorismate Synthate*) by binding to their promoters, negatively regulating SA biosynthesis and powdery mildew resistance [90]. Furthermore, FHY3 and FAR1 also regulate the concentration of SA and enhance resistance to plant disease [202].

In addition to participating in the regulation of the JA and SA signaling pathways, the light signaling pathway can also regulate plant tolerance to biotic stresses through other means (Figure 6). PIF3 negatively regulates the expression of plant defense genes and resistance to *Pseudomonas syringae* (DC3000) in Arabidopsis, and the phosphorylation of PIF3 is required for the negative regulation of plant immunity [203]. In sweet potato, *IbPIF3.1* was strongly induced by *Fusarium oxysporum* f. sp. *batatas* (*Fob*) and stem nematodes; ectopic expression of *IbPIF3.1* in tobacco significantly enhanced Fusarium wilt tolerance of transgenic plants [204].

## 5. Concluding Remarks and Future Perspectives

In the past few decades, researchers have conducted in-depth and comprehensive studies of the light signaling pathways and photomorphogenesis with the aid of the model plant Arabidopsis, which have advanced the understanding of the regulation of growth and development by the light signaling pathway. Phytochromes are important photoreceptors for plants to sense changing light in the external environment. Phytochromes are not only able to perceive diurnal and seasonal changes but also respond to extreme environmental stresses, which should not be overlooked. The role of phytochrome and phytochrome signaling pathways in diverse stresses has been studied in rice, wheat, tomato, and other crops, mainly through interactions between phytochrome signaling and hormone signaling pathways. However, the function identification and molecular regulation mechanisms of the key regulating factors in crop phytochrome signaling pathways still need further study, and the practice of achieving biological breeding and crop improvement through genetically improved photoreceptors and the light signaling pathway components needs further exploration and perfection. We believe that would be an economically efficient strategy to guide breeding and production practices by exploring the regulatory network between light signals and abiotic or biotic stresses. For example, we can regulate seed germination, chlorophyll synthesis, and removal of ROS under high-temperature stress through phyB. We can also improve crop tolerance to salt stress by modifying PIF genes to regulate salt-stress-related genes, ABA enrichment, and ROS clearance.

## Figures and Tables

**Figure 1 ijms-24-13201-f001:**
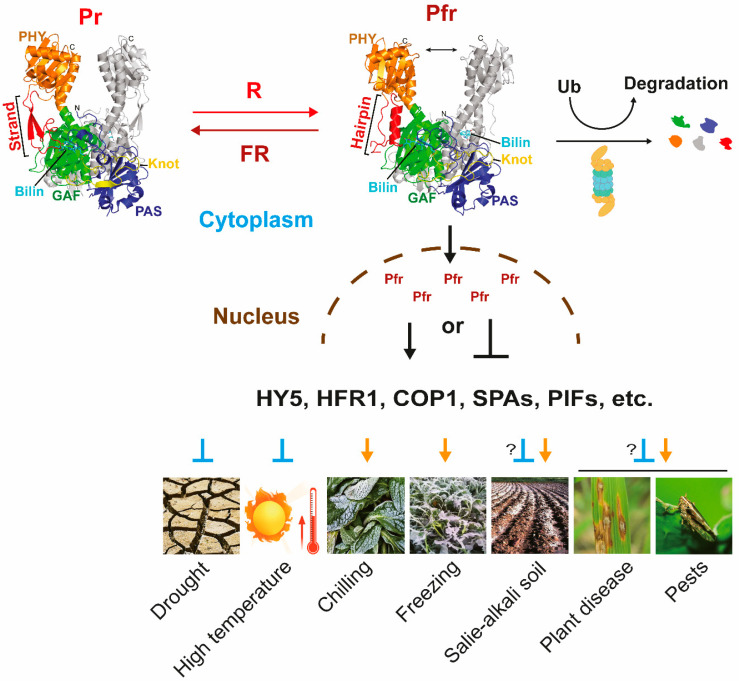
Phytochrome signaling pathway in adaptation to diverse stresses. The transition between the activated (Pfr) and inactivated (Pr) forms of the phytochromes can be realized through changes in external light quality (R/FR), in which Pfr can enter the nucleus to activate or inhibit HY5 (Elongated hypocotyl 5), HFR1 (Hypocotyl in far-red 1), COP1 (Constitutively photomorphogenic1), SPAs (Suppressor of phyA), PIFs (Phytochrome interacting factors), etc., to cope with plant responses to drought, high temperature, chilling, freezing, saline-alkali soil, plant disease, and pests.

**Figure 2 ijms-24-13201-f002:**
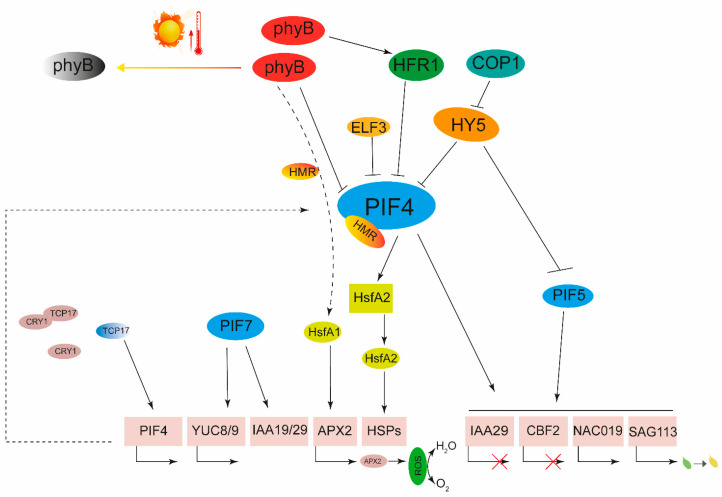
Phytochrome signaling in adaptation to high temperature. The PIFs play a large role in the phyB-dependent responses to high-temperature stress. PhyB may also directly act on HsfA1 to enhance heat resistance.

**Figure 3 ijms-24-13201-f003:**
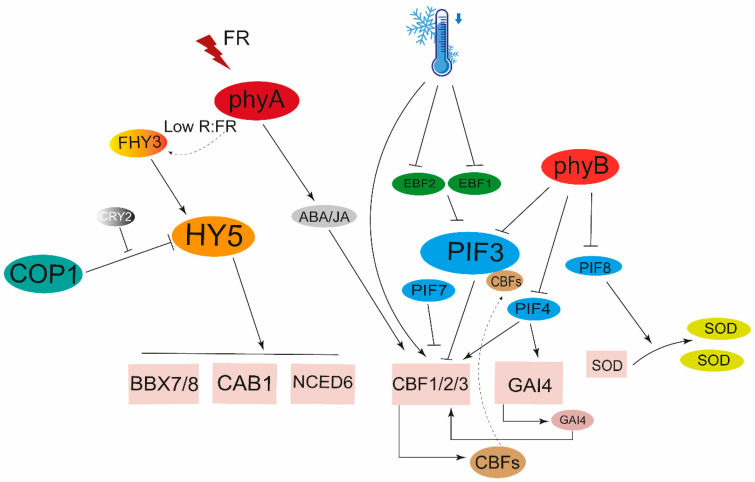
Phytochrome signaling in adaptation to low temperature. Under low-temperature conditions, plants mainly regulate the CBF through PIF3, PIF7, and PIF4 to affect the cold resistance. Of course, in addition, HY5 and phyA can also enhance the cold resistance of plants under cold conditions, while phyB acts as a negative regulator for cold in plants.

**Figure 4 ijms-24-13201-f004:**
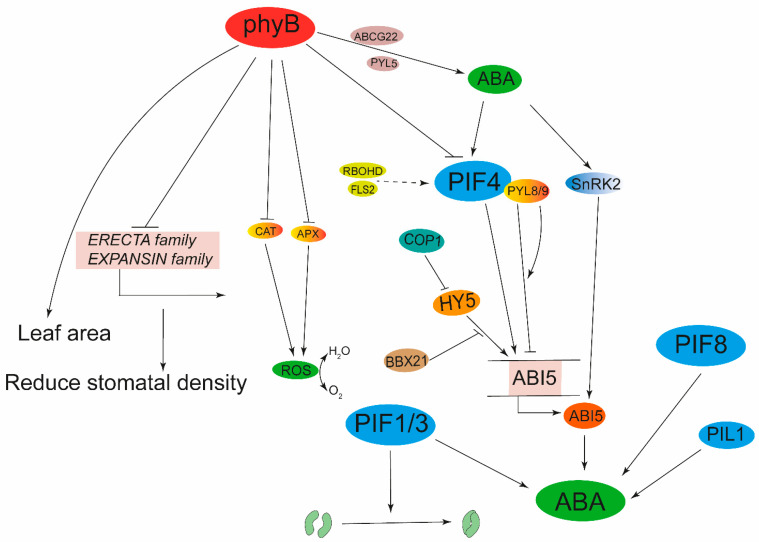
Phytochrome signaling in adaptation to drought stress. When plants were exposed to drought stress, PIF4, PIF8, PIL1, HY5, and PIF1/3 improved drought tolerance by increasing the ABA content and regulating stomatal closure, respectively. In contrast, phyB negatively regulated plant drought tolerance by inhibiting the activities of PIF4, CAT/APX, the expression of *ERECTA* family and *EXPANSIN* family genes, and leaf area.

**Figure 5 ijms-24-13201-f005:**
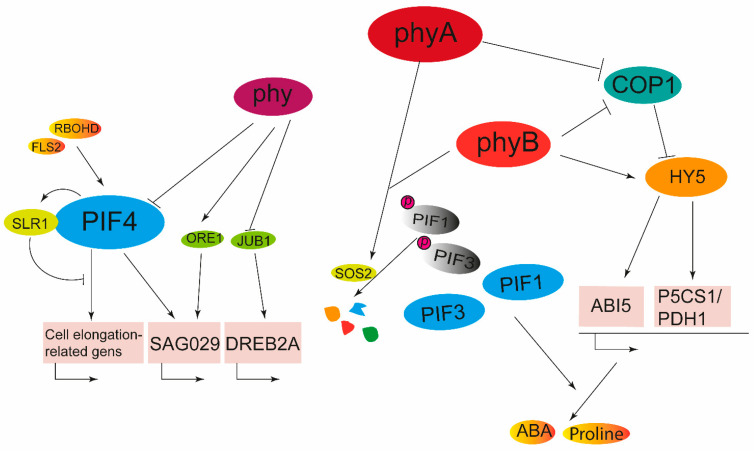
Phytochrome signaling in adaptation to salt stress. When plants face salt stress, PIF1, PIF3, and PIF4 positively regulate the expression of salt tolerance-related genes to enhance plant salt tolerance. In addition, phyB enhanced plant salt tolerance by increasing the content of HY5. RBOHD: Respiratory Burst Oxidase Protein D; FLS2: Flagellin Sensing 2; SLR1: Slender Rice 1; ORE1: Oresara 1; JUB1: Jungbrunnen 1.

**Figure 6 ijms-24-13201-f006:**
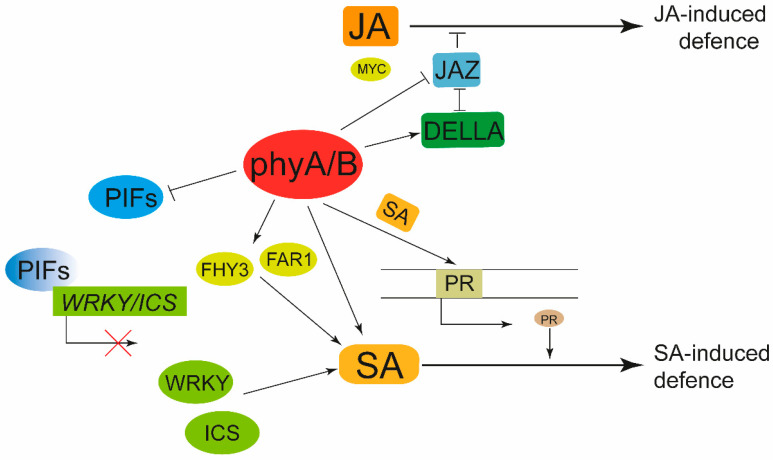
Phytochrome signaling in adaptation to biotic stresses. In the face of biotic stress, phyA/B can promote the accumulation of SA by positively promoting FHY3 and FAR1 and inhibiting PIFs. At the same time, SA can actively regulate the expression of PR and jointly initiate SA-induced defence. In addition, phyA/B can activate JA-induced defence by directly inhibiting jasmonate ZIM-domani (JAZ) or by inhibiting JAZ via DELLA.

## Data Availability

The data presented in this study are available upon request from the corresponding author.

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
