# Peer review of "Functions of Plant Phytochrome Signaling Pathways in Adaptation to Diverse Stresses"

_ijms, 2023, doi:10.3390/ijms241713201_

Round 1
Reviewer 1 Report
In the present review paper, the authors presented a list of different earlier works dealing with phytochromes and stress tolerance in plants. The topic is indeed interesting for wide range of researchers, because plant stress tolerance is one of the mostly investigated area worldwide. The paper is basically fine; however, I have a few problems with it. In a good review paper the readers expect more than simple collection of other works. Therefore Figure 1 is not informative in its present form. It is true that the ohytochrome system has influence (direct or indirect) on various signalling routes. But how do they regulate the different acclimation processes, leading to different stress tolerance mechanisms? Do they induce cold and heat tolerance in the same time, for example (for this, see a very similar review in the same journal (https://doi.org/10.3390/ijms22168602).
The auhors should clearly demonstrate the different signalling routs leading to different kind of tolerance mechanisms.
About biotic stress: this cannot also be generalised in this way. First of all, JA and SA signalling pathways may also contribute to abiotic stress tolerance. If Phytochrome system has an influence on these, why are they also mentioned at the abiotic part?
So Figure 1 can only be an introductory figure, and further explaing parts should be created forcusing on the different routes.
Future aspects: it is written "I believe that would be an economically efficient strategy to guide breeding and production practices by exploring the regulatory network between light signals and abiotic or biotic stresses through the study and understanding of phytochrome signaling pathways". In principle this is true, but very general. First of all, who is "I"? there are many authors, why not "we"? Then why did not you propose genes, pathways, directly related to Phy signalling, which may have a candidate for improving stress tolerance?
Author Response
Reviewer 1:
Comments and Suggestions for Authors
In the present review paper, the authors presented a list of different earlier works dealing with phytochromes and stress tolerance in plants. The topic is indeed interesting for wide range of researchers, because plant stress tolerance is one of the mostly investigated area worldwide. The paper is basically fine; however, I have a few problems with it. In a good review paper, the readers expect more than simple collection of other works. Therefore Figure 1 is not informative in its present form. It is true that the phytochrome system has influence (direct or indirect) on various signalling routes. But how do they regulate the different acclimation processes, leading to different stress tolerance mechanisms? Do they induce cold and heat tolerance in the same time, for example (for this, see a very similar review in the same journal (https://doi.org/10.3390/ijms22168602).
The authors should clearly demonstrate the different signalling routs leading to different kind of tolerance mechanisms.
So Figure 1 can only be an introductory figure, and further explaing parts should be created forcusing on the different routes.
R: Thank you very much for the reviewer's suggestions. In the new version, we have made modifications to the issues pointed out by the reviewer and improved the charts to more clearly demonstrate the ways and mechanisms by which the phytochrome signaling pathways regulate diverse stresses in plants using different figures (see Figure 2-6).
About biotic stress: this cannot also be generalised in this way. First of all, JA and SA signalling pathways may also contribute to abiotic stress tolerance. If Phytochrome system has an influence on these, why are they also mentioned at the abiotic part?
R: Thank you very much for the reviewer's suggestions. We have made modifications to this in section abstract.
Future aspects: it is written "I believe that would be an economically efficient strategy to guide breeding and production practices by exploring the regulatory network between light signals and abiotic or biotic stresses through the study and understanding of phytochrome signaling pathways". In principle this is true, but very general. First of all, who is "I"? there are many authors, why not "we"? Then why did not you propose genes, pathways, directly related to Phy signalling, which may have a candidate for improving stress tolerance?
R: Thank you very much for the reviewer's suggestions. In the new version, we have made modifications to the issues pointed out by the reviewer.

Reviewer 2 Report
Comments to the MS of Qiu et al. “Functions of Plant Phytochrome Signaling Pathway in Adaptation to Diverse Stresses
The above topic is an important and consideration of the authors has a number of positive points. For example, authors discussion much on PIFs. Also, I see many refs to the topic. It is OK. However, this field is really difficult for discussion and likely therefore, some points are lost. I think that, some my comments help to improve this review.
Major comments
1 In section phytochrome signaling pathway I see information mainly on PIFs. One can and needed to add an information about positive factors of hotomorphogenesis: HY5, HYH (HY5-homolog), HFR1 (long hypocotyl in far-red), and LAF1 (long after far-red light 1). Likely, one can discuss about B-box domain proteins (BBXs) and COIP1/SPA complex. HY5 is an important regulator of plant response to stressful factors.
2. HY5 is a main regulator of gene expression in Arabidopsis, at the center of several transcriptional networks (Bellegarde et al. .HNI9 and HY5 maintain ROS homeostasis under high nitrogen provision in Arabidopsis). For example, he HY5-PIF regulatory complex coordinates light and temperature control of photosynthetic gene transcription (Gabriela Toledo-Ortiz et al. 2014). One should tell about that.
3. I do not see the section on a role of phytochrome signaling in plant response to high intensity light (for example, Pavel Pashkovskiy et al. 2021).
4. There are many works devoted to a role of phytochromes and their signaling components in response of photosynthetic apparatus to different stresses.
However, there is no discussion on that. For example, Kreslavski et al. 2018, review; Pashkovskiy et al. 2021). Please, discuss this field too.
5. It would be good also see in the text some mechanisms which are present in some review papers and how they are linked to gene regulation of main components of phy signaling such as HY5,.HFR1, COP1 (see, for example Carvalho et al. 2011; 2016; Kreslavski et al. 2018). Maybe there is a sense to do a separate section devoted to these mechanisms. For example, for response to cold stress it is important the unsaturated / saturated fatty acid ratio in membranes and phyB ca affect the ration etc. One more Info for authors: respiratory burst oxidase homolog (RBOH) proteins and phyB interact during stress against ROS (Fishman et al. 2021).
6. Fig. 1. It is unclear which components are positive factors and which are negative ones. One should improve that.
Minor comments
1.I recommend to write more on a role of PIF8. For example it is known that PIF8: a new player in far-red light signaling ( Oh et al. 2020).
2. One should point out that PIFs are a family of basic helix-loop-helix (bHLH) transcription factors that have many roles.
3.Please, explain more why phyB is thermoregulator.
4. On fig. 1 I see HFR1but there is no discussion about this signaling component.
5. Some refs should be added to table 1.
6. Some abbreviations re absent. For example, FHY3
English
Sometimes singular and plural are confused. For example (Line 119), it is written: Phytochrome A-E are well known as a photoreceptor.
References recommended for authors
Toledo-Ortiz G, Johansson H, Lee KP, Bou-Torrent J, Stewart K, Steel G, et al. (2014) The HY5-PIF Regulatory Module Coordinates Light and Temperature Control of Photosynthetic Gene Transcription. PLoS Genet 10(6): e1004416. https://doi.org/10.1371/journal.pgen.1004416
Pavel Pashkovskiy, et al. Effect of high-intensity light on the photosynthetic activity, pigment content and expression of light-dependent genes of photomorphogenetic Solanum lycopersicum hp mutants. Plant Physiol. Biochem, 167, 2021, 91-100.
Jeonghwa Oh, Eunae Park, Kijong Song, Gabyong Bae, and Giltsu Choi. (2020). PHYTOCHROME INTERACTING FACTOR 8 Inhibits Phytochrome A-mediated Far-red Light Responses in Arabidopsis. Plant Cell; DOI: https://doi.org/10.1105/tpc.19.00515
Kreslavski V.D., Los D.A., F.-J. Schmitt, Zharmukhamedov S.K., Kuznetsov Vl.V., Allakhverdiev S.I. The impact of the phytochrome system on photosynthetic processes in higher plants. BBA. Bioenergetic. 2018.
R.F. Carvalho, L.R. Moda, G.P. Silva, M.A. Gavassi, R.M. Prado, Nutrition in tomato (Solanum lycopersicum L) as affected by light: revealing a new role of phytochrome, Australian, J. Crop. Sci. 10 (3) (2016) 331–335.
R.F. Carvalho, M.L. Campos, R.A. Azevedo, The role of phytochrome in stress tolerance, J. Integr. Plant Biol. 53 (2011) 920–929.
Fichman et al. 2021. Phytochrome B regulates reactive oxygen signaling during abiotic and biotic stress in plants. New Phytologist doi: 10.1111/nph.18626
Conclusion. The MS can be reconsidered again after major revision.

Comments to the MS of Qiu et al. “Functions of Plant Phytochrome Signaling Pathway in Adaptation to Diverse Stresses
The above topic is an important and consideration of the authors has a number of positive points. For example, authors discussion much on PIFs. Also, I see many refs to the topic. It is OK. However, this field is really difficult for discussion and likely therefore, some points are lost. I think that, some my comments help to improve this review.
Major comments
1 In section phytochrome signaling pathway I see information mainly on PIFs. One can and needed to add an information about positive factors of hotomorphogenesis: HY5, HYH (HY5-homolog), HFR1 (long hypocotyl in far-red), and LAF1 (long after far-red light 1). Likely, one can discuss about B-box domain proteins (BBXs) and COIP1/SPA complex. HY5 is an important regulator of plant response to stressful factors.
2. HY5 is a main regulator of gene expression in Arabidopsis, at the center of several transcriptional networks (Bellegarde et al. .HNI9 and HY5 maintain ROS homeostasis under high nitrogen provision in Arabidopsis). For example, he HY5-PIF regulatory complex coordinates light and temperature control of photosynthetic gene transcription (Gabriela Toledo-Ortiz et al. 2014). One should tell about that.
3. I do not see the section on a role of phytochrome signaling in plant response to high intensity light (for example, Pavel Pashkovskiy et al. 2021).
4. There are many works devoted to a role of phytochromes and their signaling components in response of photosynthetic apparatus to different stresses.
However, there is no discussion on that. For example, Kreslavski et al. 2018, review; Pashkovskiy et al. 2021). Please, discuss this field too.
5. It would be good also see in the text some mechanisms which are present in some review papers and how they are linked to gene regulation of main components of phy signaling such as HY5,.HFR1, COP1 (see, for example Carvalho et al. 2011; 2016; Kreslavski et al. 2018). Maybe there is a sense to do a separate section devoted to these mechanisms. For example, for response to cold stress it is important the unsaturated / saturated fatty acid ratio in membranes and phyB ca affect the ration etc. One more Info for authors: respiratory burst oxidase homolog (RBOH) proteins and phyB interact during stress against ROS (Fishman et al. 2021).
6. Fig. 1. It is unclear which components are positive factors and which are negative ones. One should improve that.
Minor comments
1.I recommend to write more on a role of PIF8. For example it is known that PIF8: a new player in far-red light signaling ( Oh et al. 2020).
2. One should point out that PIFs are a family of basic helix-loop-helix (bHLH) transcription factors that have many roles.
3.Please, explain more why phyB is thermoregulator.
4. On fig. 1 I see HFR1but there is no discussion about this signaling component.
5. Some refs should be added to table 1.
6. Some abbreviations re absent. For example, FHY3
English
Sometimes singular and plural are confused. For example (Line 119), it is written: Phytochrome A-E are well known as a photoreceptor.
References recommended for authors
Toledo-Ortiz G, Johansson H, Lee KP, Bou-Torrent J, Stewart K, Steel G, et al. (2014) The HY5-PIF Regulatory Module Coordinates Light and Temperature Control of Photosynthetic Gene Transcription. PLoS Genet 10(6): e1004416. https://doi.org/10.1371/journal.pgen.1004416
Pavel Pashkovskiy, et al. Effect of high-intensity light on the photosynthetic activity, pigment content and expression of light-dependent genes of photomorphogenetic Solanum lycopersicum hp mutants. Plant Physiol. Biochem, 167, 2021, 91-100.
Jeonghwa Oh, Eunae Park, Kijong Song, Gabyong Bae, and Giltsu Choi. (2020). PHYTOCHROME INTERACTING FACTOR 8 Inhibits Phytochrome A-mediated Far-red Light Responses in Arabidopsis. Plant Cell; DOI: https://doi.org/10.1105/tpc.19.00515
Kreslavski V.D., Los D.A., F.-J. Schmitt, Zharmukhamedov S.K., Kuznetsov Vl.V., Allakhverdiev S.I. The impact of the phytochrome system on photosynthetic processes in higher plants. BBA. Bioenergetic. 2018.
R.F. Carvalho, L.R. Moda, G.P. Silva, M.A. Gavassi, R.M. Prado, Nutrition in tomato (Solanum lycopersicum L) as affected by light: revealing a new role of phytochrome, Australian, J. Crop. Sci. 10 (3) (2016) 331–335.
R.F. Carvalho, M.L. Campos, R.A. Azevedo, The role of phytochrome in stress tolerance, J. Integr. Plant Biol. 53 (2011) 920–929.
Fichman et al. 2021. Phytochrome B regulates reactive oxygen signaling during abiotic and biotic stress in plants. New Phytologist doi: 10.1111/nph.18626
Conclusion. The MS can be reconsidered again after major revision.
Author Response
Reviewer 2:
Comments to the MS of Qiu et al. “Functions of Plant Phytochrome Signaling Pathway in Adaptation to Diverse Stresses
The above topic is an important and consideration of the authors has a number of positive points. For example, authors discussion much on PIFs. Also, I see many refs to the topic. It is OK. However, this field is really difficult for discussion and likely therefore, some points are lost. I think that, some my comments help to improve this review.
Major comments
1 In section phytochrome signaling pathway I see information mainly on PIFs. One can and needed to add an information about positive factors of hotomorphogenesis: HY5, HYH (HY5-homolog), HFR1 (long hypocotyl in far-red), and LAF1 (long after far-red light 1). Likely, one can discuss about B-box domain proteins (BBXs) and COIP1/SPA complex. HY5 is an important regulator of plant response to stressful factors.
R: Thank you very much. We have added descriptions of the two important regulatory factors, COP1 and HY5, to the section phytochrome signaling pathway.
- HY5is a main regulator of gene expression in Arabidopsis, at the center of several transcriptional networks (Bellegarde et al., HNI9 and HY5 maintain ROS homeostasis under high nitrogen provision in Arabidopsis). For example, he HY5-PIF regulatory complex coordinates light and temperature control of photosynthetic gene transcription (Gabriela Toledo-Ortizet al. 2014). One should tell about that.
R: Thanks for your suggestion. In the newly submitted version, we have added relevant discussions (section 3.1).
- I do not see the section on a role of phytochrome signaling in plant response to high intensity light (for example, Pavel Pashkovskiy et al. 2021).
R: Thank you very much for the reviewer's suggestions. We have added a discussion on this aspect in the newly submitted version.
- There are many works devoted to a role of phytochromes and their signaling components in response of photosynthetic apparatus to different stresses. However, there is no discussion on that. For example, Kreslavski et al. 2018, review; Pashkovskiy et al. 2021). Please, discuss this field too.
R: Thanks for your suggestion. We have added some discussion on this aspect in the new version (section 3.3 and 3.5).
- It would be good also see in the text some mechanisms which are present in some review papers and how they are linked to gene regulation of main components of phy signaling such as HY5, HFR1, COP1 (see, for example Carvalho et al. 2011; 2016; Kreslavski et al. 2018). Maybe there is a sense to do a separate section devoted to these mechanisms. For example, for response to cold stress it is important the unsaturated / saturated fatty acid ratio in membranes and phyB ca affect the ration etc. One more Info for authors: respiratory burst oxidase homolog (RBOH) proteins and phyB interact during stress against ROS (Fishman et al. 2021).
R: Thank you very much for the reviewer's suggestion. Based on existing articles and our review, we have constructed Figure 2-Figure 6 to better assist readers in understanding the pathways and mechanisms of phytochrome signaling pathways regulating abiotic and biotic stress.
- Fig. 1. It is unclear which components are positive factors and which are negative ones. One should improve that.
R: Thank you. In the newly submitted version, we have made new modifications and improvements to the charts.
Minor comments
1.I recommend to write more on a role of PIF8. For example it is known that PIF8: a new player in far-red light signaling ( Oh et al. 2020).
R: Thank you for your suggestion. We have added information about PIF8 in sections 2, 3.2, and 4 of the article.
- One should point out that PIFs are a family of basic helix-loop-helix (bHLH) transcription factors that have many roles.
R: thank you. We have added the information in the new version.
3.Please, explain more why phyB is thermoregulator.
R: Thank you very much. We have revised the relevant discussion in section 3.1 of the article.
- On fig. 1 I see HFR1but there is no discussion about this signaling component.
R: Thank you for your suggestion. HFR1 is involved in high-temperature and salt stress in plants, and we have added discussions in the corresponding sections.
- Some refs should be added to table 1.
R: Thanks. We have reorganized the Table 1.
- Some abbreviations re absent. For example, FHY3
R: Thank you very much. We have checked and completed the relevant information on abbreviations in the text.
English
Sometimes singular and plural are confused. For example (Line 119), it is written: Phytochrome A-E are well known as a photoreceptor.
R: Thank you very much. We have rechecked the entire text and corrected any errors.
References recommended for authors
Toledo-Ortiz G, Johansson H, Lee KP, Bou-Torrent J, Stewart K, Steel G, et al. (2014) The HY5-PIF Regulatory Module Coordinates Light and Temperature Control of Photosynthetic Gene Transcription. PLoS Genet 10(6): e1004416. https://doi.org/10.1371/journal.pgen.1004416
Pavel Pashkovskiy, et al. Effect of high-intensity light on the photosynthetic activity, pigment content and expression of light-dependent genes of photomorphogenetic Solanum lycopersicum hp mutants. Plant Physiol. Biochem, 167, 2021, 91-100.
Jeonghwa Oh, Eunae Park, Kijong Song, Gabyong Bae, and Giltsu Choi. (2020). PHYTOCHROME INTERACTING FACTOR 8 Inhibits Phytochrome A-mediated Far-red Light Responses in Arabidopsis. Plant Cell; DOI: https://doi.org/10.1105/tpc.19.00515
Kreslavski V.D., Los D.A., F.-J. Schmitt, Zharmukhamedov S.K., Kuznetsov Vl.V., Allakhverdiev S.I. The impact of the phytochrome system on photosynthetic processes in higher plants. BBA. Bioenergetic. 2018.
R.F. Carvalho, L.R. Moda, G.P. Silva, M.A. Gavassi, R.M. Prado, Nutrition in tomato (Solanum lycopersicum L) as affected by light: revealing a new role of phytochrome, Australian, J. Crop. Sci. 10 (3) (2016) 331–335.
R.F. Carvalho, M.L. Campos, R.A. Azevedo, The role of phytochrome in stress tolerance, J. Integr. Plant Biol. 53 (2011) 920–929.
Fichman et al. 2021. Phytochrome B regulates reactive oxygen signaling during abiotic and biotic stress in plants. New Phytologist doi: 10.1111/nph.18626
Conclusion. The MS can be considered again after major revision.

Round 2
Reviewer 1 Report
The authors substantially improved their MS. I only have a few minor comments:
- Figure legends must be more self-explanatory. You should write a more detailed legend for each figures deonstrating the differences (for Fig 1) and the specificity (figs. 2-6) of the different stressors.
- In Fig. 1: what is the meaning of "......" under HY5?
- Aspects related to temperature changes have recently been reviewed (https://www.mdpi.com/1422-0067/22/16/8602). This work should also be cited and mentioning the recent findings in 1-2 sentences compared to that review.
Reviewer 2 Report
Second comments to the MS of Xue Qiu et al. Functions of Plant Phytochrome Signaling Pathway in Adaptation to Diverse Stresses
The MS was improved and grate work was done but there are a number of inaccuracies mainly in English. I would like to show that using some examples:
I suggest to change Abstract (see below).
Abstract (old version). Phytochromes are receptors for red light (R)/far-red light (FR), which are not only involved in regulating the growth and development of plants, but also mediated resistance to diversity stresses. Studies have revealed that the phytochrome signaling pathway plays a crucial role in enabling plants to cope with abiotic stresses such as high/low temperatures, drought, and salinity. Phytochromes and components in light signaling pathways can also respond to biotic stresses, In addition, phytochromes can trigger the activation of jasmines acid (JA) and salicylic acid (SA) defense pathways, thereby inducing plant resistance against biotic stresses such as insect pests and microbial pathogens. Given that, this paper reviews recent advances in understanding the mechanisms of phytochromes in plant resistance to adversity and discusses the importance of modulating genes involved in phytochrome signaling pathways to coordinate plant growth, development, and stress response.
![]()
Abstract (new proposed version): Phytochromes are receptors for red light (R)/far-red light (FR), which are not only involved in regulating the growth and development of plants, but also mediated resistance to various stresses. Studies have revealed that the phytochrome signaling pathway plays a crucial role in enabling plants to cope with abiotic stresses such as high/low temperatures, drought, high intensity light and salinity. Phytochromes and their components in light signaling pathways can also respond to biotic stresses caused by insect pests and microbial pathogens, thereby inducing plant resistance against them. Given that, this paper reviews recent advances in understanding the mechanisms of action of phytochromes in plant resistance to adversity and discusses the importance of modulating genes involved in phytochrome signaling pathways to coordinate plant growth, development, and stress responses.
Line 43. It is written: Not only that, phyB……What is that? It is unclear.
Line 55-56. It is written: which mainly absorbs UV-A in the range of 320-400 nm and blue light (B) between 400-500 nm to mediate blue light and ultraviolet-A (UV-A). One should improve that and write: which mainly absorbs UV-A in the range of 320-400 nm and blue light (B) between 400-500 nm to mediate blue light and ultraviolet-A (UV-A) induced plant responses.
Line 317-318. It is written: …by altering gene expression and photosynthetic apparatus. Likely,
activity of photosynthetic apparatus.
Line 353. It is written: mutation of PHYB can enhances…... One should write: phyB mutation can enhance…
In addition. I do not see the detailed description of captions to figures.

Second comments to the MS of Xue Qiu et al. Functions of Plant Phytochrome Signaling Pathway in Adaptation to Diverse Stresses
The MS was improved and grate work was done but there are a number of inaccuracies mainly in English. I would like to show that using some examples:
I suggest to change Abstract (see below).
Abstract (old version). Phytochromes are receptors for red light (R)/far-red light (FR), which are not only involved in regulating the growth and development of plants, but also mediated resistance to diversity stresses. Studies have revealed that the phytochrome signaling pathway plays a crucial role in enabling plants to cope with abiotic stresses such as high/low temperatures, drought, and salinity. Phytochromes and components in light signaling pathways can also respond to biotic stresses, In addition, phytochromes can trigger the activation of jasmines acid (JA) and salicylic acid (SA) defense pathways, thereby inducing plant resistance against biotic stresses such as insect pests and microbial pathogens. Given that, this paper reviews recent advances in understanding the mechanisms of phytochromes in plant resistance to adversity and discusses the importance of modulating genes involved in phytochrome signaling pathways to coordinate plant growth, development, and stress response.
![]()
Abstract (new proposed version): Phytochromes are receptors for red light (R)/far-red light (FR), which are not only involved in regulating the growth and development of plants, but also mediated resistance to various stresses. Studies have revealed that the phytochrome signaling pathway plays a crucial role in enabling plants to cope with abiotic stresses such as high/low temperatures, drought, high intensity light and salinity. Phytochromes and their components in light signaling pathways can also respond to biotic stresses caused by insect pests and microbial pathogens, thereby inducing plant resistance against them. Given that, this paper reviews recent advances in understanding the mechanisms of action of phytochromes in plant resistance to adversity and discusses the importance of modulating genes involved in phytochrome signaling pathways to coordinate plant growth, development, and stress responses.
Line 43. It is written: Not only that, phyB……What is that? It is unclear.
Line 55-56. It is written: which mainly absorbs UV-A in the range of 320-400 nm and blue light (B) between 400-500 nm to mediate blue light and ultraviolet-A (UV-A). One should improve that and write: which mainly absorbs UV-A in the range of 320-400 nm and blue light (B) between 400-500 nm to mediate blue light and ultraviolet-A (UV-A) induced plant responses.
Line 317-318. It is written: …by altering gene expression and photosynthetic apparatus. Likely,
activity of photosynthetic apparatus.
Line 353. It is written: mutation of PHYB can enhances…... One should write: phyB mutation can enhance…
In addition. I do not see the detailed description of captions to figures.
